# Verification of the Interaction Target Protein of the Effector ApCE22 of *Arthrinium phaeospermum* in *Bambusa pervariabilis × Dendrocalamopsis grandis*

**DOI:** 10.3390/biom13040590

**Published:** 2023-03-25

**Authors:** Xinmei Fang, Peng Yan, Adjei Mark Owusu, Tianhui Zhu, Shujiang Li

**Affiliations:** 1College of Forestry, Sichuan Agricultural University, Chengdu 611130, China; 2College of Life Sciences, Neijiang Normal University, Neijiang 641100, China; 3National Forestry and Grassland Administration, Key Laboratory of Forest Resources Conservation and Ecological Safety on the Upper Reaches of the Yangtze River, Chengdu 611130, China

**Keywords:** effector protein ApCE22, interaction protein, B2 protein, chaperone protein DnaJ A6 chloroplastic protein

## Abstract

The study of interaction proteins of the pathogen *A. phaeospermum* effector protein is an important means to analyze the disease-resistance mechanism of *Bambusa pervariabilis × Dendrocalamopsis grandis* shoot blight. To obtain the proteins interacting with the effector ApCE22 of *A. phaeospermum*, 27 proteins interacting with the effector ApCE22 were initially identified via a yeast two-hybrid assay, of which four interaction proteins were obtained after one-to-one validation. The B2 protein and the chaperone protein DnaJ chloroplast protein were then verified to interact with the ApCE22 effector protein by bimolecular fluorescence complementation and GST pull-down methods. Advanced structure prediction showed that the B2 protein contained the DCD functional domain related to plant development and cell death, and the DnaJ protein contained the DnaJ domain related to stress resistance. The results showed that both the B2 protein and DnaJ protein in *B. pervariabilis × D. grandis* were the target interaction proteins of the ApCE22 effector of *A. phaeospermum* and related to the stress resistance of the host *B. pervariabilis × D. grandis*. The successful identification of the pathogen effector interaction target protein in *B. pervariabilis × D. grandis* plays an important role in the mechanism of pathogen–host interaction, thus providing a theoretical basis for the control of *B. pervariabilis × D. grandis* shoot blight.

## 1. Introduction

In the process of invading the host, a fungal pathogen will be resisted by the host plant’s autoimmune system, which involves two immune responses. The first is a broad-spectrum immune response (PAMPs trigger immune) triggered by pathogen-associated molecular patterns (PAMPs). To suppress PTI (PAMPs trigger immune), the pathogen will secrete effectors as an invasion weapon. However, the immune receptor R proteins in the plant will recognize effectors, causing a hypersensitive reaction (HR) and inhibiting the further infection of pathogenic fungi [1]. These effectors belong to the avirulence (*Avr*) gene, while the gene-for-gene hypothesis asserts a one-on-one relationship between the *Avr* genes and plant resistance (R) genes [2]. It has been found that many effectors can regulate the infection of the pathogen and the resistance of the host by acting on the target genes. The *Ustilago maydis* effector *Pep1* directly targets the maize peroxidase *POX12*, thereby inhibiting ROS generating capacity and early defense responses [3]. Effector *Pst18363* in *Puccinia striiformis* directly targets the Nudix hydrolase TaNUDX23 and suppresses ROS accumulation to promote pathogen infection [4]. *Pst_12806* in *P. striiformis* targets wheat *TaISP* and reduces photosynthesis, electron transport, and chloroplast-derived ROS [5]. The *Magnaporthe oryzae* effector AVR-Pii directly targets the OsNADP-ME2 in rice and inhibits the rice NADP-malic enzyme activity, resulting in the disruption of oxidative burst and host innate immunity [6]. Similarly, pathogen *A. phaeospermum* may secrete some effectors to suppress the immune response of the host *B. pervariabilis × D. grandis* during infection of the hybrid bamboo, thus aiding its own infestation.

The shoot blight caused by *A. phaeospermum* has led to a large amount of *B. pervariabilis × D. grandis* death, huge economic losses, and destruction of the ecological barrier. Most of the studies on the disease are about the pathogen *A. phaeospermum*, including the isolation and identification of the pathogen, the characteristics of disease occurrence, prevention methods, and other aspects of its physiological biology [7,8,9]. Moreover, studies on this pathogen also include bioinformatics such as whole-genome, transcriptome, protein analyses, and molecular biology analysis, such as the functional verification of virulence-related genes of the pathogen [10,11,12]. The research on the host *B. pervariabilis × D. grandis* has only involved proteome and transcriptome sequencing analysis after the plant has been infected by the pathogen [13,14,15]. However, there is no research on the interaction target proteins of *A. phaeospermum* effectors in *B. pervariabilis × D. grandis*. *A. phaeospermum* as a kind of hemibiotrophic pathogen, deploying a combination of diverse effectors, which suppress innate immunity during the biotrophic phase and elicit nonspecific cell death during the necrotrophic phase. The functions of these effectors are mainly realized by directly acting on the target genes of host plants [16]. In addition, *ApCE22* effectors in *A. phaeospermum* have been proved to have positive effects on its virulence and can inhibit the immune response of host plants [17]. Therefore, the screening and identification of *A. phaeospermum* effector ApCE22 interaction targets in host plants are particularly important in the study of *B. pervariabilis × D. grandis* shoot blight. *A. phaeospermum* effector *ApCE22* has been identified from the Dual seq sequencing data of *A. phaeospermum* and *B. pervariabilis × D. grandis*. *ApCE22* has been proved to be an effector related to the virulence of *A. phaeospermum* in previous studies, and it could participate in hydrogen peroxide metabolism and thus inhibit the PTI response of host plants [17]. However, the interaction target protein of effector ApCE22 in the host hybrid bamboo is still unclear, which greatly limits the research progress of pathogenic mechanisms and host disease-resistance mechanisms. In this study, yeast two-hybrid, two-molecule fluorescence complementation and GST pull-down techniques were used to screen and identify the interacting proteins of the pathogen effector ApCE22 in *B. pervariabilis × D. grandis*. Finally, two interacting proteins B2 and DnaJ were identified, and might play an important role in the response and disease resistance of hybrid bamboo to external stresses.

## 2. Materials and Methods

### 2.1. Microorganism, Plant and Plasmid Vectors

Microorganism: *A. phaeospermum* was stored in the China Forestry Culture Collection Center, numbered cfcc 86860 (http://www.cfcc-caf.org.cn/, accessed on 1 January 2023). The Y187 yeast strain (containing reporter genes lacZ and MEL1) and Y2H gold yeast strain (containing reporter genes AUR1-C, ADE2, HIS3, and MEL1) were obtained from OE Biotech (Shanghai, China). Agrobacterium GV3101 was obtained from TransGen Biotech (Beijing, China). *E. coli* DH5α and BL21 were obtained from GeneCreate Biological Engineering Co. Ltd. (Wuhan, China).

Plant: Two-year-old *B. pervariabilis* × *D. grandis* bamboo seedlings purchased from Shuyang Qichen Bamboo Seedling Co., Ltd. (Suqian, China) were planted in the greenhouse in Chengdu, Sichuan, China. The study area was at an altitude of 515.98 m, with annual temperatures between 6.8 °C and 26.1 °C and annual precipitation of 1300–1700 mm.

Plasmid vectors: pGBKT7-53 was used to interact with PGADT7-T as a positive control, pGBKT7-Lam was used to interact with PGADT7-T as a negative control, pGBKT7 was used to construct a bait expression vector or as a blank control, and PGADT7 was used to construct a prey expression vector. pSPYNE(R)173 and pSPYCE(M) vectors were used for the bimolecular fluorescence complementation experiment. PGEX-6P-1 and pET28a vectors were used for the GST pull-down experiment.

Culture media and reagents: SD/-Trp/-Leu/-His/-Ade/x-a-gal/AbA, SD/-Trp, SD/-Leu, SD/-Trp/-Leu, SD/-Trp/-Ade/x-a-gal/AbA and SD/-Trp/-Leu/-His/x-a-gal/AbA. Aureobasidin A (AbA).

### 2.2. Screening of the Effector ApCE22’s Interacting Protein via Yeast Two-Hybrid

#### 2.2.1. Construction of the Bait Expression Vector pGBKT7–ApCE22

The pGBKT7 vector was linearized by the BamHI (New England Biolabs, Ipswich, MA, USA) restriction endonuclease, and the effector *ApCE22* primer with the complementary sequence of BamHI enzyme site of pGBKT7 was designed by Premier 5.0 software (Appendix A). The bait gene *ApCE22* was amplified by 2× TransTaq High Fidelity PCR SuperMix (TransGen Biotech, Beijing, China). The bait vector pGBKT7–ApCE22 was obtained by overnight treatment with homologous recombinant enzyme Exnase at 16 °C. It was transformed into *E. coli* DH5α via heat shock, followed by sequencing and detection. The positive colonies were selected for preservation, and the recombinant plasmid DNA of pGBKT7–ApCE22 was extracted and retained for subsequent use.

#### 2.2.2. Detection of the Self-Activation and Toxicity of the Bait ApCE22

Y2H gold competent cells were prepared and plasmid transformed using the Yeastmaker Yeast Transformation System 2 (Clontech, Los Angeles, CA, USA). pGBKT7-ApCE22 plasmid DNA was transferred into Y2H gold yeast competent cells, and 100 μL of transformed yeast cell diluent was spread on the corresponding selective culture medium (as shown in Appendix A). The culture was placed at 30 °C to observe colony growth for 3–5 days, and the toxicity and self-activation activity of the bait ApCE22 were analyzed.

#### 2.2.3. Two-Hybrid Screening Library

The Y2H gold yeast containing the bait plasmid pGBKT7–ApCE22 was successfully transformed and cloned into 50 mL SD/-Trp liquid medium for culturing until OD_600_ = 0.8, the Y2H gold yeast was collected and the cells resuspended in 5 mL SD/-Trp liquid medium [18,19,20]. Then, 1 mL cDNA library solution (*B. pervariabilis × D. grandis* yeast library [21] infected by *A. phaeospermum*) and 5 mL pGBKT7–ApCE22 bait solution were put into a 2 L sterile flask, cultured at 30 °C and 30 rmp for 20–24 h, and put under a 10 × 40 microscope. When the hybrid solution appeared to be a clover-shaped complex, the cells were collected using 0.9% NaCl (50 μg/mL Kanmycin, kana). The cell solution was diluted 100, 1000, and 10,000 times, and 100 μL of each gradient was taken to spread on SD/-Trp, SD/-Leu, SD/-Trp/-Leu, SD/-Trp/-Ade/x-a-gal/AbA, and SD/-Trp/-Leu/-His/x-a-gal/AbA plates, respectively. The solution was incubated at 30 °C for 3–5 days, then the number of colonies were counted and the mating efficiency was calculated. The blue clones on the primary screen plate were transferred to the SD/-Trp/-Leu/-His/-Ade/x-a-gal/AbA plate with high screening intensity for further screening. Sanger sequencing and one-to-one verification were performed on the positive clones of the SD/-Trp/-Leu/-His/-Ade/x-a-gal/AbA plates.

#### 2.2.4. One-to-One Interaction Verification between the Prey Plasmid and Bait Plasmid

Primers with complementary sequences of the BamHI enzyme sites of pGBKT7 and pGADT7 vectors were designed for the interaction genes and effector *ApCE22* (Appendix A). The restriction endonuclease BamHI (New England Biolabs, Ipswich, Massachusetts, USA) was used to linearize both the pGBKT7 and pGADT7 vectors. In addition, 2× HiFi PCR SuperMix (TransGen Biotech, Beijing, China) was used to amplify the interaction genes with *ApCE22*. The interaction genes and linearized vectors were cut and recycled for backup, pGBKT7 was connected with the effector ApCE22, and pGADT7 was connected with interacting genes B2, myb, MRH1, and DnaJ, respectively. They were each transferred into *E. coli* DH5α competent cells and plasmid DNA was extracted for use. Y2H gold competent cells were prepared according to the Yeastmaker Yeast Transformation System 2 (Clontech, USA), and 10 μL pre-denatured carrier DNA, 100ng pGBKT7–ApCE22 plasmid DNA, and 200ng prey plasmid DNA were added with 300 μL of 1xPEG/LiAC. This was placed in a 30 °C water bath, and the 20 μL DMSO that was added 30 min later was mixed evenly, heated for 15 min at 42 °C, and then the cells were collected. They were resuspended in YPD plus and incubated for 1 h. The colonies were collected, resuspended in 0.9% NaCl, and used to spread on SD/-Trp/-Leu/-His/-Ade/X-a-gal/AbA plates. The colonies were cultured at 30 °C for 3–5 days for observation. At the same time, the vector of bait and prey were exchanged and verified again.

### 2.3. Verification of the Interacting Proteins via Bimolecular Fluorescent Complementation Assay

#### 2.3.1. Construction of the Bimolecular Fluorescent Complementary Vectors pSPYNE (R) 173-ApCE22 and pSPYCE(M)-Prey

Premier 5.0 software was used to design the primer of the effector *ApCE22* with the complementary sequence of the BamHI digestion site in the pSPYNE(R)173 vector, as well as to design the primer of the interaction genes B2 and DnaJ with the complementary sequence of the BamHI digestion site in the pSPYCE(M) vector (Appendix A). The restriction endonuclease BamHI (NEB) was used to linearize both the pSPYNE(R)173 and pSPYCE(M) vectors. Then, 2×HiFi PCR SuperMix (TransGen Biotech, Beijing, China) was used to amplify the interaction gene fragments with connectors. The interaction gene fragments and linearized vectors were cut and recycled for backup. After cloning ApCE22 into the pSPYNE (R) 173 vector, and cloning B2, DnaJ into the pSPSYCE (M) vector, respectively, they were transferred into *E. coli* DH5α susceptibility. The positive colonies recombinant plasmids pSPYNE(R)173–ApCE22, pSPYCE(M)–B2, and pSPYCE(M)–DnaJ were each detected by the primer ApCE22-F/R, B2-F/R and DnaJ-F/R, respectively. Colonies with correct sequencing were selected for storage. At the same time, the vector of bait and prey were exchanged and verified again.

#### 2.3.2. Bimolecular Fluorescence Verification of the ApCE22 and Interacting Proteins in Tobacco

Tobaccos were cultivated for 3–4 weeks under 25 °C light, 16 h darkness, and 8 h of alternation. Then, we transformed the constructed BiFC recombinant plasmid (pSPYNE(R)173-ApCE22, pSPYCE(M)-B2, and pSPYCE(M)-DnaJ) into *Agrobacterium* GV3101. The pSPYNE(R)173–ApCE22 plasmid and pSPYCE(M)–B2 plasmid were mixed in equal volumes to obtain a mixed solution ApCE22-B2. The pSPYNE(R)173–ApCE22 plasmid and pSPYCE(M)–DnaJ plasmid were mixed in equal volume to obtain mixed solution ApCE22-DnaJ. The mixtures ApCE22-B2 and ApCE22-DnaJ were inoculated into tobacco leaves with syringes without needles. The infected tobaccos were cultured at 25 °C for 2 days. Fluorescence imaging was performed on the inoculated area of tobacco leaves with a laser confocal microscope. At the same time, the vector of bait and prey were exchanged and verified again. The genes bZIP63 and bZIP1 (Arabidopsis) were used as the positive control, and the three treatment groups pSPYNE (R) 173-ApCE22 and pSPYCE (M), pSPYCE (M)-B2 and pSPYNE (R) 173, and pSPYCE (M)-DnaJ and pSPYNE (R) 173 were used as negative controls. See Table 1 for the experimental settings.

### 2.4. Verification of the Interacting Proteins via GST Pull-Down Assay

#### 2.4.1. Construction of the PGEX-6P-1 and pET28a Expression Recombinant Vectors

Bioinformatics analysis was performed on the effector *ApCE22* and the ORF sequence of the interaction gene obtained from yeast two-hybrid and bimolecular fluorescence complementation, using ProtParam (http://web.expasy.org/protparam/ accessed on 1 September 2022) to predict hydrophobicity and other properties of the protein. Moreover, TMHMM 2.0 (http://www.cbs.dtu.dk/services/TMHMM-2.0/, accessed on 1 October 2022) was used to predict the transmembrane region. The online software SignalP 4.1 (http://www.cbs.dtu.dk/services/SignalP/, accessed on 1 September 2022) was used to predict the protein signal peptide. PGEX-6P-1 and pET28a were linearized using the restriction endonuclease BamHI (New England Biolabs, Ipswich, MA, USA). Primers with the complementary sequence of the BamHI digestion site in the PGEX-6P-1 vector and the complementary sequence of the BamHI digestion site in the pET28a vector were designed (Appendix A). Then, 2× TransTaq High Fidelity PCR SuperMix (TransGen Biotech, Beijing, China) was used to amplify the interaction genes *B2* and *DnaJ*. The PGEX-6P-1–ApCE22 decoy protein vector was constructed by using homologous recombinant enzyme Exnase. pET28a–B2 and pET28a–DnaJ capture protein vectors were stored for future use.

#### 2.4.2. Expression and Purification of the Fusion Protein

The constructed PGEX-6P-1-GST–ApCE22, pET28a-His–B2, and pET28a-His–DnaJ fusion protein prokaryotic expression vectors were all transformed into *E. coli* BL21 competent cells. After shake culturing at 37 °C at 225 rpm/min, the OD_600_ value was 0.6. A concentration of 0.1 mM IPTG was added to induce expression. After incubation at 16 °C for 16h, the cells were collected in 40 mL pre-cold GST balance solution, and the cells were ultrasonically broken (the parameters were set to 200 W power, 2.5 s ultrasound, a 5 s pause, and 80 cycles), and the supernatant and sediment collected. A small amount of supernatant and sediment were used for SDS-PAGE detection, and the remaining supernatant and sediment were retained at 4 °C for subsequent use.

#### 2.4.3. Pull-Down Verification

A total of 200 μL of 50% glutathione agarose resin homogenate and 500 μg of GST-ApCE22 were used as the experimental group, and 500 μg GST and 200 μL 50% glutathione agarose resin homogenate were used as the control group. Then, 500 μg His-B2 and His-DnaJ were added to the control group and experimental group, respectively. The mixture was overturned at 4 °C for overnight incubation. The washing process was repeated three times with 1 mL of pre-cooled PBST, and 80 μL of RIPA buffer cell lysate and 20 μL 6× load buffer were added, mixed well, and boiled for 5–10 min. Finally, the supernatant obtained in the previous step was separated by SDS-PAGE electrophoresis, and the protein bands were transferred to the PVDF membrane by the wet transfer method for Western blot detection.

#### 2.4.4. Modeling and Phylogenetic Analysis

The software TBtools (v1.09876) was used to build the rootless phylogenetic tree of the target protein, and the bootstrap number was 5000. Predictprotein (https://www.predictprotein.org/, accessed on 1 February 2023) was used for secondary structure prediction of the protein encoded by the gene. Online software SWISS-MODEL (http://swissmodel.expasy.org/interactive, accessed on 1 March 2023) was used to predict the tertiary structure of the protein [22]. Additionally, we used InterPro (https://www.ebi.ac.uk/interpro/, accessed on 1 March 2023) to predict the functional domains and analyze the protein’s putative functions.

## 3. Results

### 3.1. Screening of the Interaction Proteins of Effector ApCE22

The pGBKT7 and pGADT7 vectors were linearized by the BamHI enzyme. The linearized pGBKT7 vector was ligated into effector ApCE22. The positive colony was obtained after *E. coli* transformation, and the pGBKT7–ApCE22 plasmid DNA was extracted. The clone of the pGBKT7-53 and pGADT7-T positive control reaction grew in SD/-Leu/-Trp/X-α-gal and SD/-Leu/-Trp/-His/-Ade/X-α-gal/AbA medium, and the clone grew and turned blue. It indicated that the pGBKT7-53 and pGADT7-T vectors were successfully expressed, and the positive control experiment was successful. In addition, yeast colonies of the negative controls pGBKT7-Lam and pGADT7-T grew on the SD/-Leu/-Trp/X-α-gal medium, indicating that pGBKT7-Lam and pGADT7-T were normally expressed, while there were no clones on the SD/-Leu/-Trp/-His/-Ade/ X-α-gal/AbA medium, indicating the negative control experiment was successful. In addition, the reaction results of the pGBKT7–ApCE22 bait and pGADT7-T are shown in Appendix A, indicating that pGBKT7–ApCE22 had slight self-activation, but activation at SD/-Leu/-Trp/-His/-Ade/X-α-gal/AbA could be inhibited and had no toxicity (Appendix A). Thus, SD/-Leu/-Trp/-His/-Ade/X-α-gal/AbA provides the necessary conditions for subsequent screening experiments. After screening the interaction library of ApCE22 using the mating method, 52 blue clones of the pGBKT7–ApCE22 interaction protein grew on the screening plate of the SD/-Leu/-Trp/-His/-Ade/X-α-gal/AbA medium and were transferred to SD/-Leu/-Trp/-His/-Ade/X-α-gal/AbA for another screening, where 27 blue clones were obtained (Figure 1). However, multiple colonies turning blue in the same plate might have an effect on the color of the neighboring negative colonies. Therefore, the blue-positive colonies obtained from the second screening would be only half of the blue-positive colonies obtained from the first screening. The 27 positive clones were sequenced, and the sequencing results were aligned to 23 genes in the NCBI database (Table 2).

### 3.2. One-to-One Verification of Effector ApCE22 and the Interacting Proteins

After one-to-one interaction verification between bait vector pGBKT7–ApCE22 and prey vector, four prey genes interacting with the bait vector were obtained. The annotation results of the interaction gene’s functions were myb-related protein P, chaperone protein DnaJ A6 chloroplastic, B2 protein, and putative LRR receiver-like serine/threonine protein kinase MRH1. At the same time, the linearized pGADT7 vector was ligated into these four target genes, respectively. After exchanging the vector of the prey gene and bait gene, the verification was performed again. The result of the interaction between effector ApCE22 and the four proteins is shown in Appendix A. The interaction colonies after exchanging bait carrier and prey carrier are shown in Figure 2.

### 3.3. Verification the Interaction of ApCE22 and the Interacting Proteins via Bimolecular Fluorescence Complementary

The fluorescence complementation results of two molecules showed that there was an interaction between the effector ApCE22 and B2 protein, and the ApCE22, and chaperone DnaJ chloroplast protein. The PCR detection results of recombinant plasmid pSPYNE(R)173–ApCE22, pSPYCE(M)–B2, and pSPYCE(M)–DnaJ DNA by the ApCE22-F/R, B2-F/R, and DnaJ-F/R primers are shown in Appendix A. It was found that the effector ApCE22 produced fluorescence whether it reacted with the chaperone DnaJ chloroplast or with the B2 protein in tobacco (Figure 3), indicating that there was an interaction between the ApCE22 effector of A. phaeospermum with the DnaJ chloroplast and with the B2 protein in hybrid bamboo.

### 3.4. Pull-Down Validation of the Effector ApCE22 and the Interacting Protein

The sequence analysis results of the effector protein ApCE22 and the interacting proteins B2 and DnaJ showed that ApCE22 encoded for a peptide of 256 amino acids, and did not contain a transmembrane region (Appendix A), and that 1–17 amino acids are signal peptides (Appendix A). Moreover, the local hydrophilicity was good (Appendix A). The B2 protein contained 346 amino acids, had no transmembrane region and no signal peptide sequence, and had good local hydrophilicity (Appendix A). The DnaJ gene encoded 285 amino acids, had no transmembrane region and no signal peptide sequence, and had good local hydrophilicity (Appendix A). The electrophoretic results of the purified effector ApCE22 and the target proteins B2 and DnaJ are shown in Appendix A; the sizes are consistent with the sizes of the target proteins. Their Western blot results are shown in Appendix A. The results of the pull-down analysis of the purified ApCE22 protein with B2, and ApCE22 protein with DnaJ are shown in Figure 4, indicating that the effectors ApCE22 interacted with B2 and DnaJ in vitro, respectively.

### 3.5. Advanced Structure Prediction and Phylogenetic Analysis of ApCE22, B2 and DnaJ

The tertiary structure results of ApCE22, and the interacting proteins B2 and DnaJ are shown in Figure 5. Among them, the ApCE22 protein contained three helixes, which were located at amino acids 2–18, 121–124, and 235–255. Its tertiary structure also contained calcium ion and bromine ion ligands. The secondary structure prediction of the amino acid sequence encoded by B2 and DnaJ through the SPOMA website showed that the B2 protein contained an α-helix (25.14%), an extended strand (11.27%), and a random coil (57.23%). The DnaJ protein contained an α-helix (14.74%), an extended strand (8.68%), an extended strand (23.51%), a β-turn (6.32%), and a random coil (55.44%). The B2 protein contained a DCD domain related to plant development and cell death, which was located in the 210th to 342nd amino acids. The secondary structure of this domain was mainly composed of β composed chains, which had an α-spiral limit. In addition, the structure of the B2 protein also contained three SO4 ligands. The 43rd to 258th amino acids of the DnaJ protein was the DnaJ domain, which was composed of the C-terminal region of the DnaJ protein. The two C-terminal domains CTDI and CTDII of the protein were necessary to maintain the DnaJ domain at its specific relative position. The DnaJ protein structure also contained two zinc ligands. Using the adjacency method, the evolutionary tree was constructed based on the amino acid sequence of the B2 protein and DnaJ protein (Figure 6). The results showed that both the B2 protein and DnaJ protein were aligned to gramineous setaria plant, the B2 protein was aligned to Setaria italica, and the DnaJ protein was aligned to Setaria viridis.

## 4. Discussion

Based on yeast-two hybrid technology, 27 interaction target proteins of the effector ApCE22 were screened in the hybrid bamboo *B. pervariabilis × D. grandis*. Four interaction proteins were obtained via yeast two-hybrid, including myb-related protein P, chaperone protein DnaJ A6 chloroplastic, B2 protein, and putative LRR receiver-like serine/threonine protein kinase MRH1. Bimolecular fluorescence complementary and GST pull-down assays were used to prove that chaperone protein DnaJ A6 chloroplastic and the B2 protein interacted with the effector protein ApCE22 in vivo and in vitro. The B2 protein belongs to the DCD domain-containing protein NRP-like family and contains a development cell death (DCD) domain. This family represents a group of plant asparagine-rich proteins containing a DCD domain, some of which contain additional recognizable motifs, like the KELCH repeats or the ParB domain. They are involved in the stress signaling pathways that mediate cell death in response to endoplasmic reticulum (ER) stress and osmotic stress, and include DCD domain-containing proteins NRP-A/B from soybean, NRP from Arabidopsis, and B2 protein from carrot [23,24,25]. Among them, DCD domain-containing protein NRP in soybean is induced under ER stress as a component of the early pathogen response. In Arabidopsis, DCD domain-containing protein NRP contributes to the initial phase of responses to abiotic and biotic stress signals. It binds the phytochrome-associated protein phosphatase FYPP3 and facilitates FYPP3 degradation to promote abscisic acid (ABA) response [26,27]. It has also been found that the over-expression of the B2 protein, which contains the DCD domain in *Arabidopsis thaliana*, leads to hypersensitivity [28]. The DCD domain is a conserved domain in proteins involved in plant development and programmed cell death, and, coincidentally, the interaction target B2 protein of effector ApCE22 also contains a DCD domain. It is speculated that the target protein B2 can interact with effector ApCE22 during *A. phaeospermum* invasion, thus leading to the programmed death of host plant cells. Although a dead cell on its own might already stop the growth of biotrophic pathogens [29], the hybrid bamboo shoot blight Pathogen *A. phaeospermum*, as a kind of hemibiotrophic pathogen, can still grow and develop in the necrotrophic stage and infect the host. This also shows that the B2 protein could recognize the effector ApCE22 of *A. phaeospermum* and cause the programmed cell death of hybrid bamboo, but it does not necessarily prevent the infection of *A. phaeospermum.*

Another protein that has been proven to interact with the effector protein ApCE22 is the Chaperone protein DnaJ A6 chloroplastic, which is a molecular chaperone that widely exists in plant cells and belongs to the Hsp40 homologous superfamily. The protein contains a DnaJ domain, which can not only combine with the folded polypeptide chain to prevent its aggregation but also stimulate the ATPase activity of the helper protein DnaK so that the helper proteins DnaK and DnaJ form a complex to respond to external stress [30,31]. The chaperone protein has a precise coordination effect on biological cell cycle entry, and the cell cycle mechanism is an important part of interfering with plant growth at the cell level [32]. For example, the Ydj1 chaperone and nuclear accumulation of the G1 cyclin Cln3 are inversely dependent on growth rate and readily respond to changes in protein synthesis and stress conditions that alter protein-folding requirements [33]. In addition, Ydj1, the same Hsp40 chaperone involved in releasing Cln3 from the ER, has also been shown to play an important role in the degradation mechanisms of this G1 cyclin [34,35]. Similar to DnaJ protein, Ydj1 is a kind of J-domain chaperone protein. It plays a role in protein homeostasis and releasing G1 cyclin Cln3 from the endoplasmic reticulum to trigger cell cycle entry [36]. Heat shock protein Hsp26 also participates in the UPR response provoked by ER stress [37].

At present, the DnaJ/Hsp40 family protein has been found to participate in the development and adversity stress of plants, including abiotic stress such as heat tolerance, salt stress, drought stress, and biological stress such as improving plant disease resistance [38,39,40,41]. The silencing cotton DnaJ/Hsp40 family protein GhDNAJ1 enhances cotton susceptibility to *Verticillium dahlia* [42]. Overexpression of a tomato chloroplast-targeted *DnaJ* gene enhances resistance to *Pseudomonas solanacearumm* [40]. The association between the capsid protein (CP) and host DnaJ-like proteins (HSP40) is essential during potato virus Y infection [43]. Similarly, the tobacco type I DnaJ protein NbMIP1 is required for tobacco innate immunity to the mosaic virus infection [44]. In addition, the soybean type III DnaJ protein GmHSP40 is essential for disease resistance [45]. What is more, the DnaJ protein in rice has been proven to be an important regulator of chloroplast growth and development, and the DnaJ protein in *Arabidopsis* has also been found to be essential for chloroplast growth [38,46,47]. 

In conclusion, this study first screened 27 interacting proteins from a hybrid bamboo yeast library using the pathogenic ApCE22 effector protein as bait, followed by one-to-one protein interaction validation to obtain two proteins, B2 and DnaJ, with true interactions with the ApCE22 effector protein. The B2 protein is mainly involved in the programmed cell death and other immune reactions of hybrid bamboo. The DnaJ protein not only plays an important role in the growth and development of chloroplast synthesis of hybrid bamboo leaves, but also plays an important role in heat tolerance, salt tolerance, drought stress, and resistance to *A. phaeospermum*. Although the DnaJ and B2 proteins have been proved to interact with the effector ApCE22, their functions have been only preliminarily predicted. Therefore, it is still necessary to identify their functions in the host *B. pervariabilis × D. grandis* through gene knockout, gene silencing, or gene expression techniques.

## Figures and Tables

**Figure 1 biomolecules-13-00590-f001:**
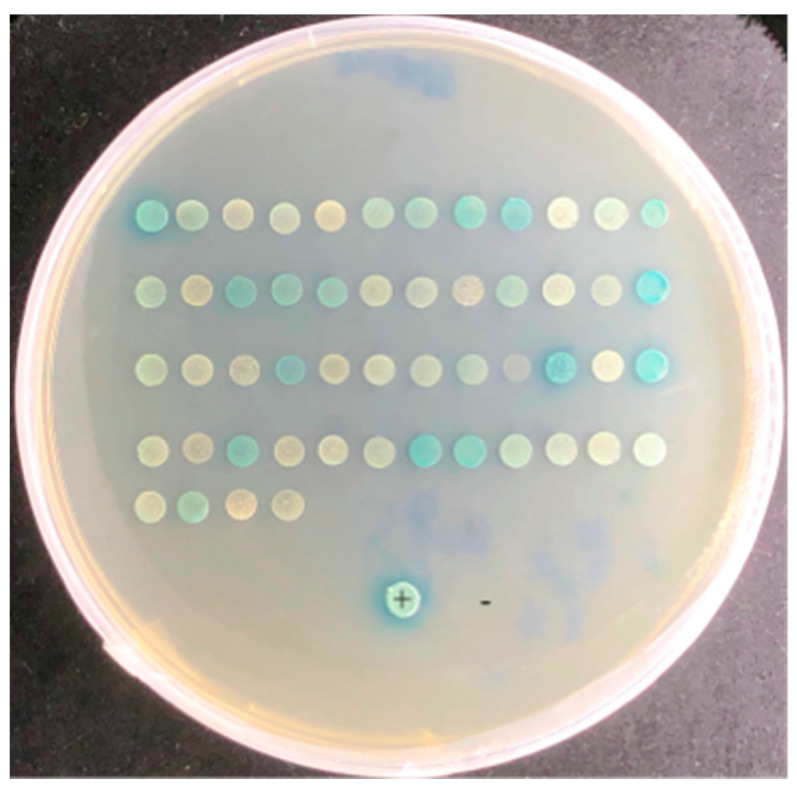
Positive clones of the pGBKT7–ApCE22 interaction gene on a double-screen plate. Note: +: stands for positive control. −: stands for negative control. The first screening step is to screen for interacting prey proteins from a yeast library of hybrid bamboo using effector ApCE22 as a bait protein, and the direct screening marker for this step is colony bluing. The second screening step is to pick out the blue colonies obtained in the first step individually and screen them again.

**Figure 2 biomolecules-13-00590-f002:**
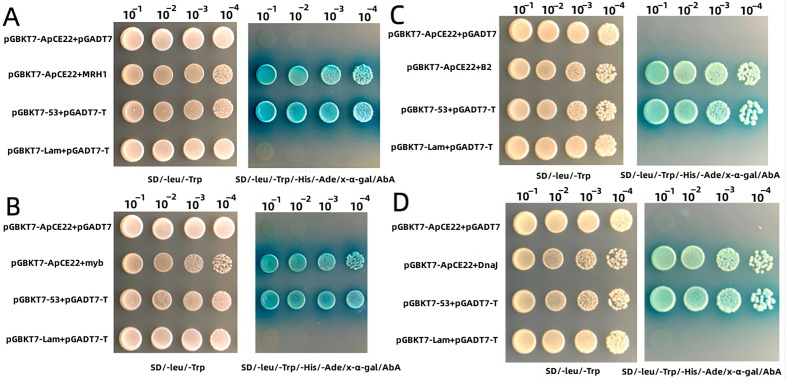
One-to-one yeast two-hybrid results of the interacting proteins and ApCE22. Note: Interaction proteins were as a prey protein, and ApCE22 protein was as a bait protein. (**A**): One-to-one yeast two-hybrid results of MRH1 proteins and ApCE22. (**B**): One-to-one yeast two-hybrid results of B2 proteins and ApCE22. (**C**): One-to-one yeast two-hybrid results of myb proteins and ApCE22. (**D**): One-to-one yeast two-hybrid results of DnaJ proteins and ApCE22.

**Figure 3 biomolecules-13-00590-f003:**
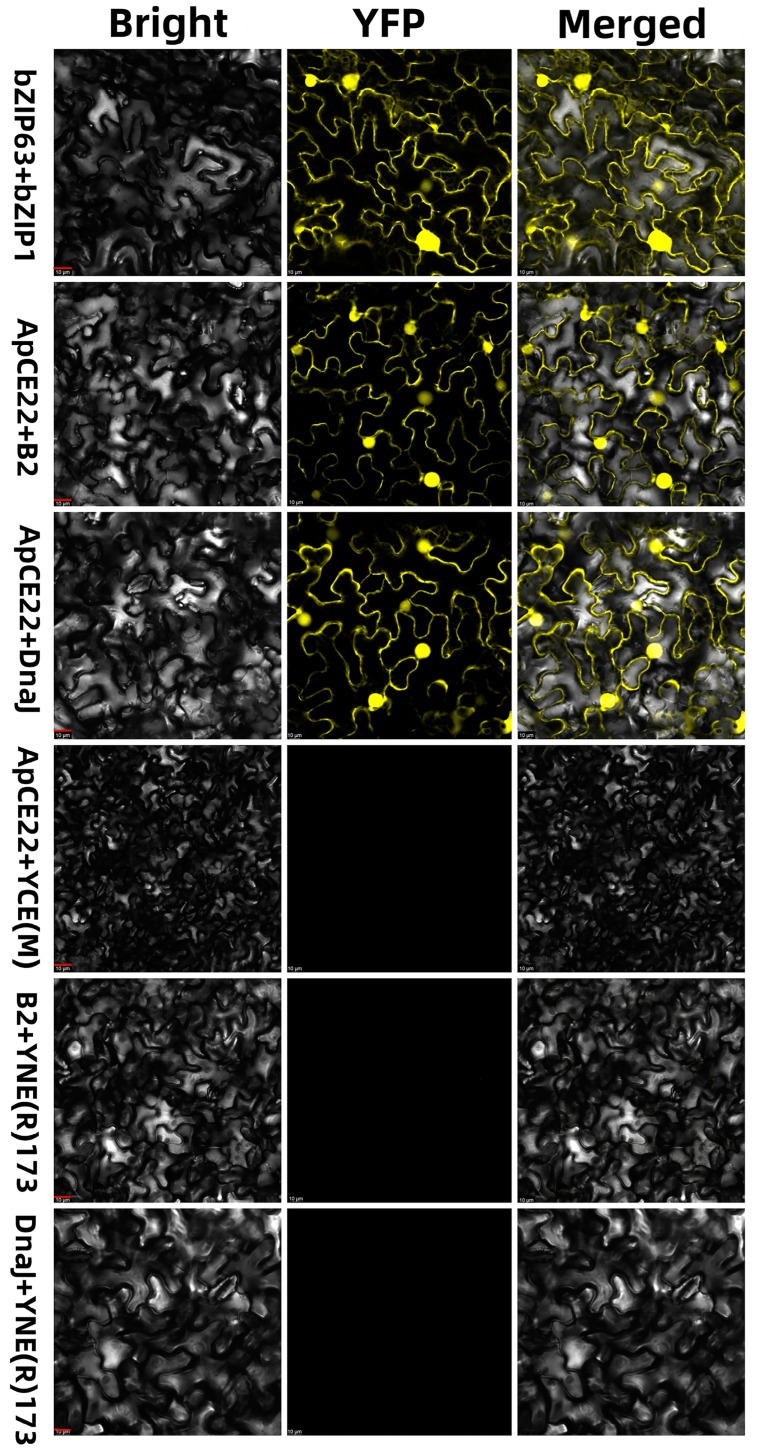
Results of the bimolecular fluorescence complementary reaction between the effector ApCE22 and the interaction proteins in tobacco. Note: The red bar indicates 10 μm. Tobacco is a recognized substitute for verifying protein interactions in hybrid bamboos that have not yet successfully established a genetic transformation system.

**Figure 4 biomolecules-13-00590-f004:**
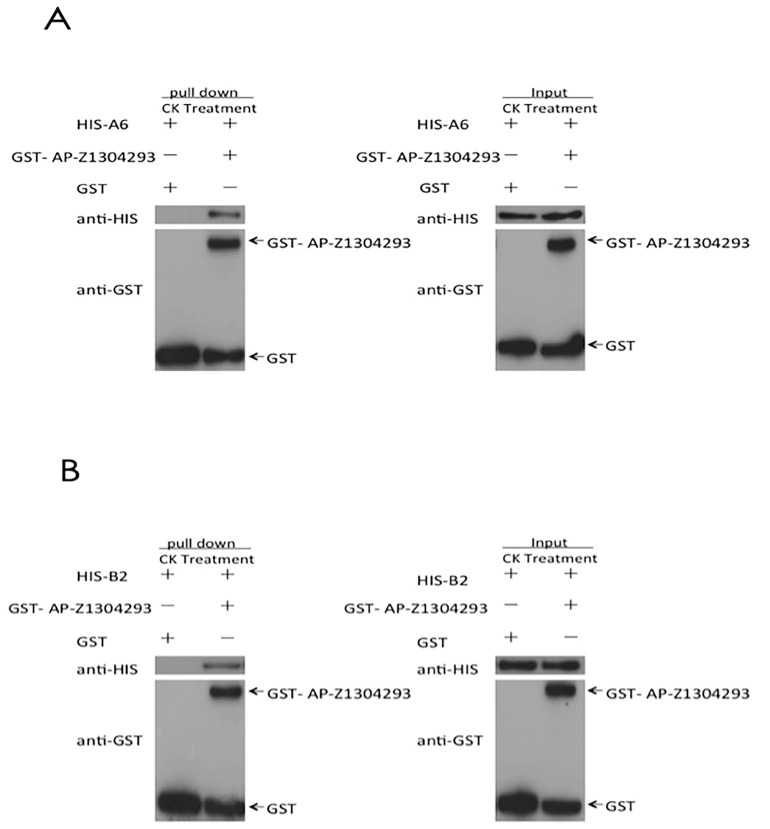
GST pull-down results of the target proteins B2 and DnaJ. Note: (**A**): interaction between ApCE22 and DnaJ. (**B**): Interaction between ApCE22 and B2. Pull-down is the protein electrophoresis result of the eluent obtained after the DnaJ protein solution passes through the GST column. Input is the protein electrophoresis result of the DnaJ protein solution without passing through the GST column.

**Figure 5 biomolecules-13-00590-f005:**
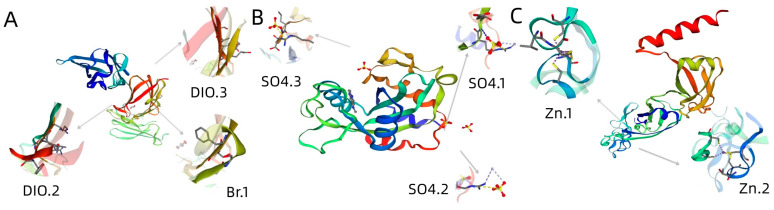
Homology modeling of effectors and interacting proteins. Note: (**A**): effector ApCE22 contains two DIO ligands and one Br ligand, where the DIO ligand can combine DIO ions, and Br ligand can combine Br ions; (**B**): B2 protein contains three SO_4_ ligands and DCD domain (amino acids 210th to 342th), where the SO_4_ ligand can be a non-polymer covalently linked to polymer or other heterogen groups; (**C**): DnaJ protein contains two Zn ligands and DnaJ domain (amino acids 43rd to 258th), where the Zn ligand can combine zinc ions.

**Figure 6 biomolecules-13-00590-f006:**
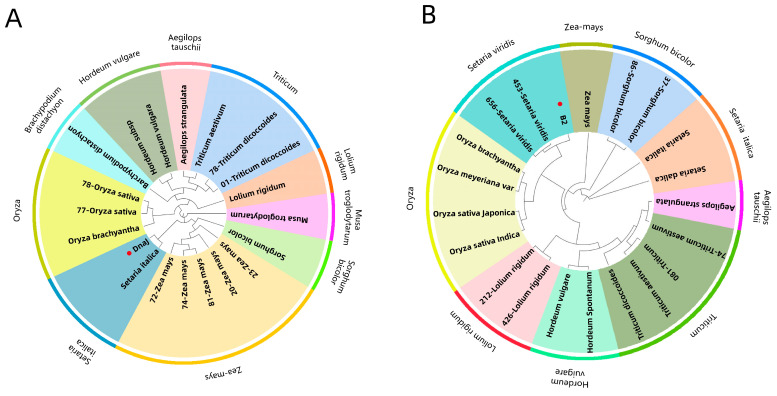
Phylogenetic trees of B2 and DnaJ proteins. Note: (**A**): the phylogenetic tree of the relationship between the DnaJ protein and Gramineae was constructed using the adjacency method (NJ). The different color plates on the circle represent different genera of Gramineae, and the solid red dots represent the DnaJ protein in hybrid bamboo. (**B**): The phylogenetic tree of the relationship between the B2 protein and Gramineae was constructed using the adjacency method (NJ). The different color plates on the circle represent different genera of Gramineae, and the solid red dots represent the B2 protein in hybrid bamboo. Because the whole genome sequence of *B. pervariabilis × D. grandis* is not yet available, the interacting proteins can only be mapped to other species.

**Table 1 biomolecules-13-00590-t001:** BiFC’s experimental setup.

Plasmid	Positive CK	Negative CK1	Negative CK2	Negative CK3	Treatment 1	Treatment 2
pSPYNE(R)173–bZIP6	+	−	−	−	−	−
pSPYCE(M)–bZIP	+	−	−	−	−	−
pSPYCE(M)–DnaJ	−	+	−	−	−	+
pSPYCE(M)–B2	−	−	+	−	+	−
pSPYNE(R)173–ApCE22	−	−	−	+	+	+

Note: + represents that the reaction system contains this kind of plasmid, − represents that the reaction system does not contain this kind of plasmid.

**Table 2 biomolecules-13-00590-t002:** Functional annotation of target gene of effector *ApCE22*.

Gene Number	Gene Name	Gene Bank
ApCE22-AD-01	hypothetical protein GQ55_2G039300 (*Panicum hallii var. hallii*)	PUZ68577.1
ApCE22-AD-03	hypothetical protein E2562_020380 (*Oryza meyeriana*)	KAF0913219.1
ApCE22-AD-04	hypothetical protein E2562_024347 (*Oryza meyeriana*)	KAF0896488.1
ApCE22-AD-05	hypothetical protein E2562_014243 (*Oryza meyeriana*)	KAF0888431.1
ApCE22-AD-06	hypothetical protein BRADI_1g16670v3 (*Brachypodium distachyon*)	PNT74528.1
ApCE22-AD-27		
ApCE22-AD-07	40S ribosomal protein S15a-5 (*Setaria italica*)	XP_004964360.1
ApCE22-AD-08	hypothetical protein E2562_030933 (*Oryza meyeriana*)	KAF0919682.1
ApCE22-AD-09	putative LRR receptor-like serine/threonine-protein kinase MRH1	KAE8770201.1
ApCE22-AD-10	B2 protein (*Brachypodium distachyon*)	XP_003569184.1
ApCE22-AD-22		
ApCE22-AD-11	hypothetical protein E2562_003088 (*Oryza meyeriana*)	KAF0921280.1
ApCE22-AD-12	myb-related protein P (*Brachypodium distachyon*)	XP_003558096.1
ApCE22-AD-23		
ApCE22-AD-13	laccase-22 (*Brachypodium distachyon*)	XP_003558760.1
ApCE22-AD-14	chaperone protein dnaJ A6 chloroplastic (Zea mays)	AQK41669.1
ApCE22-AD-21		
ApCE22-AD-15	macrodontain-1 (*Brachypodium distachyon*)	XP_003562772.1
ApCE22-AD-17	PREDICTED: Oryza sativa Japonica group histone H1, mRNA	XM_015788867.2
ApCE22-AD-18	elongation factor 1-alpha (*Dichanthelium oligosanthes*)	OEL30955.1
ApCE22-AD-19	myb-related protein Hv33 (*Setaria italica*)	XP_004961344.1
ApCE22-AD-20	hypothetical protein PFICI_14414 (*Pestalotiopsis fici W106-1*)	XP_007841186.1
ApCE22-AD-16	acidic ribosomal protein P2a-2 (*Zea mays*)	AQK92355.1
ApCE22-AD-02	hypothetical protein EE612_052460, partial (*Oryza sativa*)	KAB8113458.1
ApCE22-AD-24	Os10g0555900, partial (*Oryza sativa Japonica Group*)	BAT11994.1
ApCE22-AD-25	aldose 1-epimerase-like (*Hordeum vulgare*)	KAE8818356.1
ApCE22-AD-26	hypothetical protein EJB05_39871, partial (*Eragrostis curvula*)	TVU16314.1

## Data Availability

Not applicable.

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
