# Peer review of "Verification of the Interaction Target Protein of the Effector ApCE22 of *Arthrinium phaeospermum* in *Bambusa pervariabilis × Dendrocalamopsis grandis"

_biomolecules, 2023, doi:10.3390/biom13040590_

Round 1

Reviewer 1 Report

I think the authors did a great job regarding the research design and the methodology of the study, I have the following concerns

1- Please increase the resolution for figures 7 and 8

2- In Figure 5, please leave a space (horizontal and vertical) between panels and make sure about the intensity and contrast in all the panels

3- label for protein ladder bands are missing and necessary in figure 6

4- make sure that gene names are in italic throughout the manuscript

5- I recommend citing these references for yeast methodologies

a- Amponsah, P. S., Yahya, G., Zimmermann, J., Mai, M., Mergel, S., Mühlhaus, T., Storchova, Z., & Morgan, B. (2021). Peroxiredoxins couple metabolism and cell division in an ultradian cycle. Nature chemical biology, 17(4), 477–484. https://doi.org/10.1038/s41589-020-00728-9

b- Yahya, G., Menges, P., Amponsah, P. S., Ngandiri, D. A., Schulz, D., Wallek, A., Kulak, N., Mann, M., Cramer, P., Savage, V., Räschle, M., & Storchova, Z. (2022). Sublinear scaling of the cellular proteome with ploidy. Nature communications, 13(1), 6182. https://doi.org/10.1038/s41467-022-33904-7

c- Yahya, G., Wu, Y., Peplowska, K., Röhrl, J., Soh, Y. M., Bürmann, F., Gruber, S., & Storchova, Z. (2020). Phospho-regulation of the Shugoshin - Condensin interaction at the centromere in budding yeast. PLoS genetics, 16(8), e1008569. https://doi.org/10.1371/journal.pgen.1008569

Reviewer 2 Report

After carefully rectifying the problems listed below, the manuscript is ready for publishing.

The abstract is written haphazardly. Writers should include their noteworthy discoveries and effectively link them with the phrases they choose to relate.

The beginning portion of your essay must provide a clear hypothesis and considerably develop the second paragraph. Increase its relevance to the issue statement.

L72-76 have some redundancy.

L84-88 should be rewritten clearly.

Generally, there is information repetition that might be eliminated.

When writing scientific names, use italics.

The last paragraph of the discussion section has considerable repetition.

The conclusion section should be more specific and emphasise the current manuscript's results.

That should be redone. Overall, after integrating ideas, it may be appropriate for publishing.

Reviewer 3 Report

Dear authors,

please find my remarks and suggestions in the order of the text and not in the order of importance. Major comments will be highlighted in yellow.

Rewrite abstract

L14-15. “The study of target interaction proteins of A. phaeospermum effectors is an important means to analyze to understand the disease resistance mechanism of Bambusa pervariabilis × Dendrocalamopsis grandis shoot blight.”

L16-17. “To obtain the proteins  interacting with the effector ApCE22 of A. phaeospermum, 27 proteins interacting with the effector”

L22. “the B2 protein contained the a DCD functional domain”

L24-26. “ The results showed that both the B2 protein and DnaJ protein in B. pervariabilis × D. grandis were the target interaction proteins of the ApCE22 effector of A. phaeospermum and related to the stress resistance of the host B. pervariabilis × D. grandis.” How does this relate with the previous sentence on structure prediction?

Introduction 

L 36. “broad-spectrum immune response (PTI)” what does PTI stand for?

L53. “destroyed destruction of the construction of the ecological barrier” destruction of the construction?

L61. “the interaction target genes” ?

L85.  ” Four proteins interacting with effector ApCE22 in hybrid bamboo were obtained by yeast two-hybrid technology” it was 27 in the abstract

Material and Methods 

Verify the material and methods section. It looks like a cut and paste from the protocol.

l.137” When the hybrid solution appeared to be clover-shaped conjugates”?

Results 3.1 

l264. “linearized pGADT7 vector was ligated into target genes B2, DnaJ, ATP, and B-box, respectively.” Where do those come from? How many host interactors were tested?

L279-283. What is the difference between the two screening steps? What is the significance of losing about ½ the interactors?

 Table 2.  As the B. pervariabilis × D. grandis yeast library was used, why do all the hits map to other species?

Figure 3 should be labeled similarly to figure 4.

 Results 3.3

L351. “The gene detection results”??

L349-357. Why is the model switched to tobacco? Can the hybrid bamboo be transfected? If not, what are the indications that tobacco is a good substitute?

results 3.4.

Pull-down should be validated from plant extract or even better from infected plant extracts.

Section 3.5

how was the homology modelling performed? (not described in methods!!). There are no references on this section and it is unclear where the information come from.

From the abstract it appeared that the DCD and DNA J domains were important features but there is no mention of these domains here.

L396-397 : “both the B2 protein and DnaJ protein were aligned to gramineous plants, which were consistent with the hybrid bamboo of gramineous plants.” What conclusion do you get from this?

I suppose the alignment is on the whole protein sequence. Is there information in aligning the different domains?

Figure 7 is difficult to see and unhelpful. What are we supposed to see here?

Discussion 

L412-413. “Four interaction proteins were obtained via yeast two-hybrid,” I’m still confused by this sentence. To me., 4 interactors were validated by a subsequent Y-2-H screen. This actually points to the fact that there is no justification for choosing these particular proteins.

L432. “Similarly, the interaction target B2 432 protein of effector ApCE22 also contains a DCD domain” circular argument.

L433. “It is speculated that …” based on this study only or in others?

L442. “Another protein that has been proven to interact with the effector protein ApCE22” same as above

L471-475. “Therefore, it is speculated that the cause of the symptoms of hybrid  bamboo shoot blight is (…) causing the branches and leaves to turn yellow.” Overreach

Round 2

Reviewer 1 Report

I think the authors covered all my concerns, and I accept the manuscript for publication in its current form.

Reviewer 3 Report

Dear authors,

thank you for your efforts. I have few minor comments that could help the readability of the manuscript.

- Figure 1 and the bottom part of Figure 3 could be moved to the supplementary information as they are not required to understand the paper.

- in Figure 7, indicate what Dio, Br, So4 and Zn stand for. Alos indicate that those are the DCD and DNAJ domains.

-Include your response to my comments of l287-283 in supplementary info.

- Include "Note that because the whole genome sequence of B. pervariabilis × D. grandis is not yet available, the interacting proteins can only be mapped to other species." in the manuscript.

- Similarly, include " tobacco is a recognized substitute for verifying protein interactions in hybrid bamboos that have not yet successfully established a genetic transformation system" in the manuscript.

-modelling and phylogenetic analysis should be a separate sub-section in Methods
